# EMMA: Efficient Visual Alignment in Multi-Modal LLMs

**Sara Ghazanfari**[*],    **Alexandre Araujo**

**Prashanth Krishnamurthy**,    **Siddharth Garg**,    **Farshad Khorrami**

*Department of Electronic and Computer Engineering, New York University*

[*]*sg7457@nyu.edu*

**Reviewed on OpenReview:** *https://openreview.net/forum?id=lbrO3bGpeO*

## Abstract

Multi-modal Large Language Models (MLLMs) have recently exhibited impressive general-purpose capabilities by leveraging vision foundation models to encode the core concepts of images into representations. These are then combined with instructions and processed by the language model to generate high-quality responses. Despite significant progress in enhancing the language component, challenges persist in optimally fusing visual encodings within the language model for task-specific adaptability. Recent research has focused on improving this fusion through modality adaptation modules, but at the cost of significantly increased model complexity and training data needs. In this paper, we propose **EMMA** (**E**fficient **M**ulti-**M**odal **A**daptation), a lightweight cross-modality module designed to efficiently fuse visual and textual encodings, generating instruction-aware visual representations for the language model. Our key contributions include: (1) an efficient early fusion mechanism that integrates vision and language representations with minimal added parameters (less than 0.2% increase in model size), (2) an in-depth interpretability analysis that sheds light on the internal mechanisms of the proposed method; (3) comprehensive experiments that demonstrate notable improvements on both specialized and general benchmarks for MLLMs. Empirical results show that EMMA boosts performance across multiple tasks by up to **9.3%** while significantly improving robustness against hallucinations.

## 1 Introduction

Over the past years, Large Language Models (LLMs) have transformed natural language processing (NLP) by demonstrating exceptional abilities in understanding, generating, and reasoning with text across a wide range of tasks; from machine translation and summarization to complex problem-solving and conversational agents (Touvron et al., 2023; Zheng et al., 2023). However, many real-world applications require the ability to process more than just text, such as understanding visual content or synthesizing information from different modalities Ghazanfari et al. (2023b;a). This has led to the development of multi-modal LLMs [1], which combine the linguistic strengths of LLMs with vision foundation models, enabling cross-modal understanding and reasoning. By integrating textual and visual information, these models extend the capabilities of traditional LLMs to address tasks like image captioning, visual question answering, and text-to-image generation(Liu et al., 2024b; Alayrac et al., 2022; Achiam et al., 2023).

Current state-of-the-art multi-modal models typically rely on fixed visual feature encodings extracted from vision foundation models, which are projected into the text space and passed to the language model along with instructions (Driess et al., 2023b; Liu et al., 2024a;b). However, the static nature of these encodings, formed without considering the instruction, limits the model's ability to adapt dynamically to specific tasks or contexts. This disconnect between visual and textual components reduces flexibility, making the model less

---

[1]In this paper, we refer use multi-modality specifically for the combination of image and text modalities.

responsive to task-specific nuances. To address this, BLIP-2 (Li et al., 2023b) introduced a cross-attention-based module (called Q-former) to integrate the visual and instruction encodings, a design later adopted by others (Li et al., 2023b; Huang et al., 2023; Ye et al., 2023b). The current state-of-art model, mPLUG-Owl2 (Ye et al., 2024), introduces a *modality adaptation module* that employs attention modules to embed the two modalities into a shared semantic space and thus enhances cross-modality collaboration. mPLUG-Owl2's performance improvement comes with several limitations compared to baseline models. First, mPLUG-Owl2 leverages LLaMA's text embeddings and CLIP's vision encoder to generate the instruction and visual encodings, respectively. Therefore, the encodings are generated using two distinct models with no initial multi-modality alignment. Second, mPLUG-Owl2's modality-adaptive module introduces roughly 1B more parameters, $3\times$ more than its vision encoder. The modularity adaptation module is then trained from *scratch*, requiring 348 million image-text pairs for pertaining, $300\times$ more than the baseline. Third, the vision encoder requires training during both the pretraining and instruction-tuning stages, which increases the overall training cost and makes the vision encoder more susceptible to loss of generality. Finally, except for a few benchmarks, the model offers only marginal improvements, and in some cases, performs worse than the baseline.

The challenges outlined above led us to explore a more efficient method for modality adaptation. We hypothesized that the need for a complex module for modality adaptation arises from the fact that visual and textual encodings are produced by two entirely separate modules, trained independently. As a result, these complex modules attempt to integrate two distinct spaces, which is inherently difficult. To address this issue, we introduce **EMMA** (**E**fficient **M**ulti-**M**odal **A**daptation), which performs modality fusion through a lightweight modality adaptation mechanism. EMMA integrates CLIP's text encoder with its visual encoder and leverages the pre-trained alignment to adapt visual representations with the instruction via an efficient modality adaptation module (adding less than 0.03% parameters to the model). Our modality adaptation module generates instruction-aware visual representations by attending to more informative, instruction-related tokens, leading to improvements in MLLM-specialized and general benchmarks. A comprehensive set of experiments on benchmarks demonstrates that EMMA significantly enhances cross-modal alignment, improves performance across a range of vision-language tasks, and strengthens the robustness of MLLMs against hallucination. Our contributions can be summarized as follows:

- **Efficient Modality Adaptation**: We introduce a lightweight modality adaptation mechanism that refines visual representations with less than a 0.2% increase in model size, maintaining high efficiency without compromising performance.

- **Comprehensive Analysis of Visual Alignment**: We conduct an in-depth investigation of the Visual Alignment module to provide (1) a detailed understanding of how visual and textual tokens are integrated and (2) an analysis of how effectively the aligned visual representations attend to instructions compared to the initial raw visual encodings.

- **Extensive Empirical Evaluation**: We perform a comprehensive evaluation on both general and MLLM-specialized benchmarks, demonstrating that EMMA significantly improves cross-modal alignment, boosts task performance, and enhances the robustness of multi-modal LLMs.

- **EMMA Outperforms Larger Models**: Compared to mPLUG-Owl2, which has $50\times$ larger modality adaptation module and is trained on $300\times$ more data, EMMA outperforms on 7 of 8 benchmarks. Additionally, compared with BRAVE, which has $24\times$ larger vision encoder and is trained on $100\times$ more data, EMMA outperforms on all benchmarks.

## 2 Related Work

**Multi-modal Large Language Models (MLLMs).** In recent years, there has been significant progress in the development of multi-modal large language models (LLMs) that integrate vision and language to handle tasks requiring both modalities (Zhang et al., 2024; 2023; Wu et al., 2023; Sun et al., 2024; Alayrac et al., 2022; Lai et al., 2023; Li et al., 2023a;g;d; Lin et al., 2024; Liu et al., 2023b;c; Tian et al., 2024; Wang et al., 2024b;c; Chen et al., 2023a). By combining the language understanding of LLMs with the perceptual

| Method | Vision Encoder | Modality Adapter | LLM | Total |
|--------|---------------|------------------|-----|-------|
| **InstructBLIP** | 1.3B $_{\times 3.3}$ | 200M$_{\times 10}$ | 7B | 7.91B |
| **BLIP-2/InstructBLIP** | 1.3B$_{\times 3.3}$ | 200M$_{\times 10}$ | 13B | 14.2B |
| **Qwen-VL-Chat** | 1.9B$_{\times 6.3}$ | 80M$_{\times 4}$ | 7.7B | 9.6B |
| **BRAVE** | **7.2B**$_{\times 24}$ | 100M$_{\times 5}$ | 2.6B | 10B |
| **mPLUG-Owl2** | 0.3B$_{\times 1}$ | **1000M**$_{\times 50}$ | 7B | 8.2B |
| **EMMA** | 0.3B | 22M | 7B | 7.3B |

Table 1: Parameter sizes of EMMA compared with leading MLLMs across three components: Vision Encoder, Modality Adapter, and LLM. EMMA uses the same-sized Vision Encoder and LLM as mPLUG-Owl2, but its Modality Adapter, our main contribution, is 50× smaller.

abilities of vision foundation models, multi-modal LLMs are able to tackle a wide array of tasks that require cross-modal alignment and understanding. We can divide these models into two general categories based on the way that the two visual and textual modalities are integrated. In the first category, including LLaVA (Liu et al., 2024b;a), PaLM-E (Driess et al., 2023a), Shikra (Chen et al., 2023b), etc., the vision encodings are projected to the text space with a few linear layers, and then concatenated with the instruction tokens and passed to the LLMs. In the second category, a more complex module is used for cross-modality adaptation, where both visual and textual encodings are processed through the adaptation module. First introduced with Flamingo (Alayrac et al., 2022) and later adopted by BLIP-2 (Li et al., 2023c), InstructBLIP (Dai et al., 2024), Qwen-VL (Bai et al., 2023), mPLUG-Owl (Ye et al., 2023a), and MiniGPT-4 (Zhu et al., 2024), the use of a Q-former (a cross-attention-based module) has become a prominent technique.

**Enhancing Visual Alignment in MLLMs.** Since the emergence of multi-modal LLMs, aligning visual and textual modalities has remained a significant challenge in achieving robust and seamless integration. Previous works (Alayrac et al., 2022; Li et al., 2023c; Ye et al., 2023a;c; Kar et al., 2024) have predominantly focused on utilizing Q-formers and similar cross-attention modules as modality adaptation components to integrate visual and textual embeddings. Recent advancements in modality adaptation include mPLUG-Owl2 (Ye et al., 2023c) and BRAVE (Kar et al., 2024). mPLUG-Owl2 (Ye et al., 2023c) introduces 1B modality adaptation module that employs distinct parameters to project multiple modalities into a unified semantic space for enhanced modality adaptation. On the other hand, BRAVE (Kar et al., 2024) leverages a concatenation of various visual encodings, summing up to 7B, directly feeding into the Q-former. These modality adaptation modules rely on intricate architectures, introducing millions to billions of additional parameters to the model, which significantly increases the computational costs. Recently, proposed a Question-Aware Vision Transformer approach that integrates question-awareness directly into the vision encoder. While this design enhances task-specific performance, it compromises the generality of the visual representations. This added complexity not only demands vast amounts of training data but also imposes considerable overhead during inference. Furthermore, the complication of these systems makes it difficult to discern the primary drivers behind performance improvements — whether they stem from the model complexity, the early fusion of vision and text encodings, or the sheer volume of additional training data. This leads us to the central focus of our work: (1) addressing the inefficiencies in current multi-modal models by proposing a more streamlined approach. Our goal is to achieve an efficient early fusion of visual and textual encodings without significantly increasing the number of parameters or computational costs. (2) Conducting a thorough analysis to determine the key drivers behind performance improvements and the dynamics of the modality adaptation module to generate the multi-modal encodings. This investigation will help isolate the most effective factors and guide future advancements in multi-modal LLMs.

**Multi-modal LLM's Benchmarks.** The evaluation of multi-modal LLMs relies on a mix of traditional academic benchmarks and newer ones tailored to instruction-following MLLMs. Established benchmarks like VQA-v2 (Goyal et al., 2017) and GQA Hudson & Manning (2019) gauge a model's ability to interpret visuals through open-ended, short-answer questions. ScienceQA (Lu et al., 2022b) tests zero-shot generalization

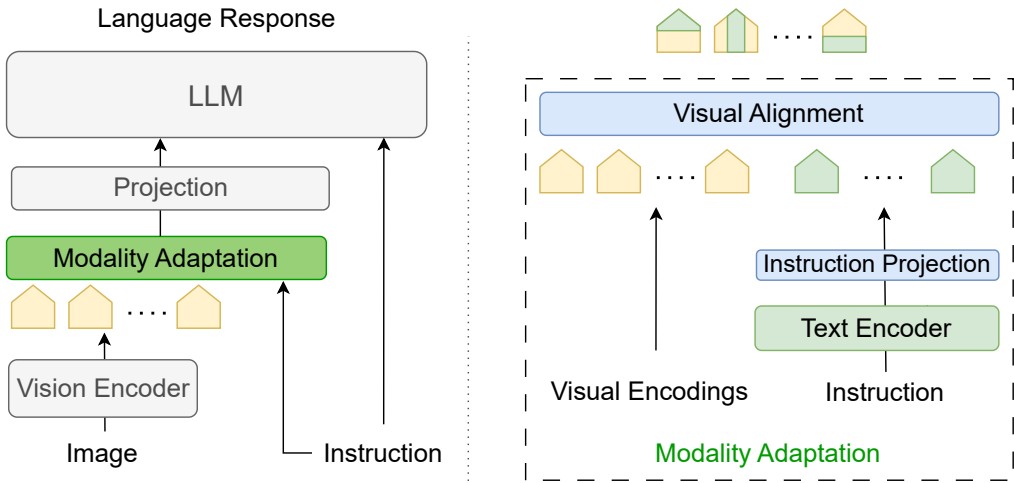

Figure 1: This figure presents the **EMMA's Architecture** where Modality Adaptation is introduced to the standard MLLM architecture. The Modality Adaptation module consists of the Text Encoder, Instruction Projection, and Visual alignment modules. EMMA enhances the multi-modality alignment by incorporating the instruction encodings generated by the CLIP's Text Encoder, which are then projected to the joint space by the Instruction Projection module. The concatenating visual and textual encodings are then passed through the Visual Alignment module, resulting in multi-modal, instruction-aware representations.

in scientific question answering, while VizWiz (Gurari et al., 2018) offers real-world images captured by visually impaired users, challenging models with poor framing, blur, and other quality issues typical of non-professional photos. Additionally, newer benchmarks target instruction-following MLLMs. MathVista (Lu et al., 2023) introduces diverse challenges from mathematical and visual tasks. MMMU (Yue et al., 2024) evaluates multi-modal models on a broad range of college-level tasks requiring deep subject knowledge and reasoning. To assess multi-image reasoning, MUIRBENCH (Wang et al., 2024a) provides a comprehensive benchmark with 12 diverse tasks to evaluate the multi-image understanding abilities of MLLMs.

For general robustness, MMBench (Liu et al., 2023a) offers a multiple-choice visual question answering benchmark in both English and Chinese, suggesting shuffling the choices to test the model's robustness to the order of options. MMVP (Tong et al., 2024) evaluates robustness by identifying similar images with minute differences and manually pinpointing the visual details the CLIP vision encoder overlooks, which leads to incorrect responses from MLLMs. For hallucination, POPE (Li et al., 2023f) examines the extent of hallucinations across three COCO subsets, and AMBER (Wang et al., 2023b), a multi-dimensional benchmark, evaluates generative and discriminative tasks. Additionally, three benchmarks were utilized to assess the proposed method's robustness against hallucination: FOIL (Shekhar et al., 2017), MMRel (Nie et al., 2024), and R-Bench (Wu et al., 2024).

## 3 Efficient Visual Alignment in Multi-Modal LLMs

In this section, we explain our proposed method which addresses the inefficiencies in current multi-modal models by an efficient early fusion of visual and textual encodings without significantly increasing the number of parameters or computational costs. Moreover, a detailed interpretability analysis is offered to provide insights into the internal mechanisms of the proposed method.

### 3.1 EMMA: Efficient Multi-Modal Adaptation

Recent progress in multi-modal models has been largely driven by the robust reasoning capabilities of large language models. Therefore, a persistent challenge is to effectively align these two modalities to ensure

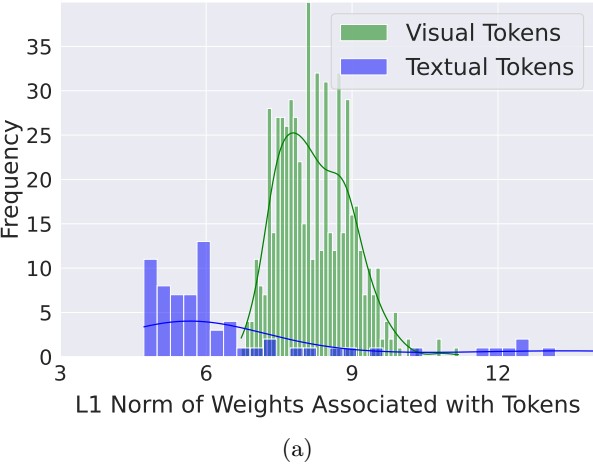 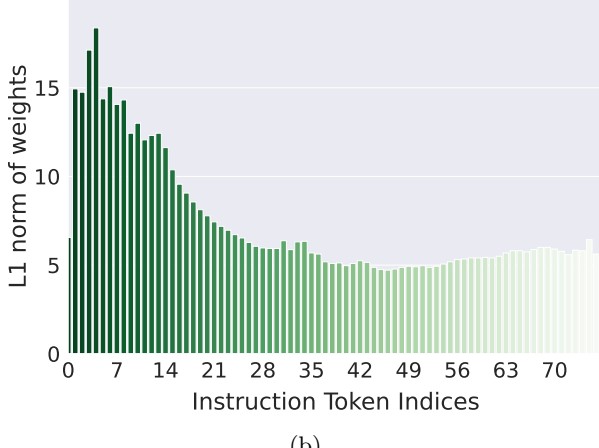

(a)             (b)

Figure 2: To evaluate the **contribution of visual and textual tokens to the Aligned Representations**, we analyze the weight matrix of the Visual Alignment module, which consists of a linear layer. **Left**: Fig. 2a presents the $\ell_1$ norms of the weights for each token, revealing that (1) visual tokens have a stronger influence on the aligned representations and (2) the impact of textual tokens varies, prompting further investigation in Fig. 2b to identify which specific textual tokens contribute more significantly to the final representations. **Right**: The bar plot reveals that the early tokens are assigned higher weights in the Visual Alignment module, placing greater emphasis on them.

seamless fusion and task-specific adaptability. Current approaches often rely on complex cross-modality modules, which introduce a significant number of parameters and thus require large amounts of training data, as shown in Table 1. We hypothesize that the need for a complex modality adaptation module arises from the fact that visual and textual encodings are produced by two independently trained components that are themselves unaligned. This is precisely the case for mPLUG-Owl2, which uses CLIP as its vision encoder and LLaMA's text embeddings for its text encoder. As a result, the multi-modality module must, in addition to incorporating text information into the visual embedding, also align the two embeddings.

To address this issue, we propose a simple but surprisingly effective idea—we use both CLIP's vision encoder *and* its text encoder in our multi-modality alignment module. By directly incorporating CLIP's text encoder into its visual encodings, Since CLIP's vision and text encoders were originally jointly trained, multi-modal adaptation is inherently embedded in their encodings, making its text encoder an ideal choice for encoding instructions. The strong, inherent alignment between the two modalities allows for seamless integration, minimizing the need for complex cross-modal modules or extensive training to achieve alignment. Furthermore, CLIP has demonstrated strong performance across diverse tasks, making it a reliable foundation for multi-modal applications.

We refer to our proposed architecture as **EMMA— E**fficient **M**ulti-**M**odal **A**daptation. Figure 1 illustrates EMMA's architecture. The left side highlights the high-level structure, where the standard modules of a multi-modal LLM are shaded in gray, while EMMA's newly introduced Modality Adaptation module is shown in green. On the right, the details of the Modality Adaptation module are depicted. More specifically, let $v(\cdot)$ and $t(\cdot)$ represent the vision encoder and text encoder of CLIP, respectively. The visual encodings are defined as $\mathbf{v} = v(\mathbf{x}) \in \mathbb{R}^{n \times d}$, where $\mathbf{x}$ is the input image, and the text encodings as $\mathbf{t} = t(\mathbf{i}) \in \mathbb{R}^{m \times d'}$, where $\mathbf{i}$ is the instruction. Later, the instruction encodings are then processed through the Instruction Projection module $p : \mathbb{R}^{d'} \to \mathbb{R}^d$, which maps the instruction tokens to the same dimensional space as the visual tokens. Once the visual and textual representations are generated, we introduce early fusion via a lightweight module called the Visual Alignment module. This component, consisting of a simple linear layer, combines visual and textual tokens to create the model's multi-modal encoding. The Visual Alignment module, which consists of a linear layer, forms the core of our proposed method. Let $f : \mathbb{R}^{(n+m) \times d} \to \mathbb{R}^{n \times d}$ represent the Visual Alignment module, where $n$ and $m$ denote the number of visual and textual tokens, respectively. The Visual Alignment module takes the concatenation of the visual and textual tokens as input to generate $n$ refined

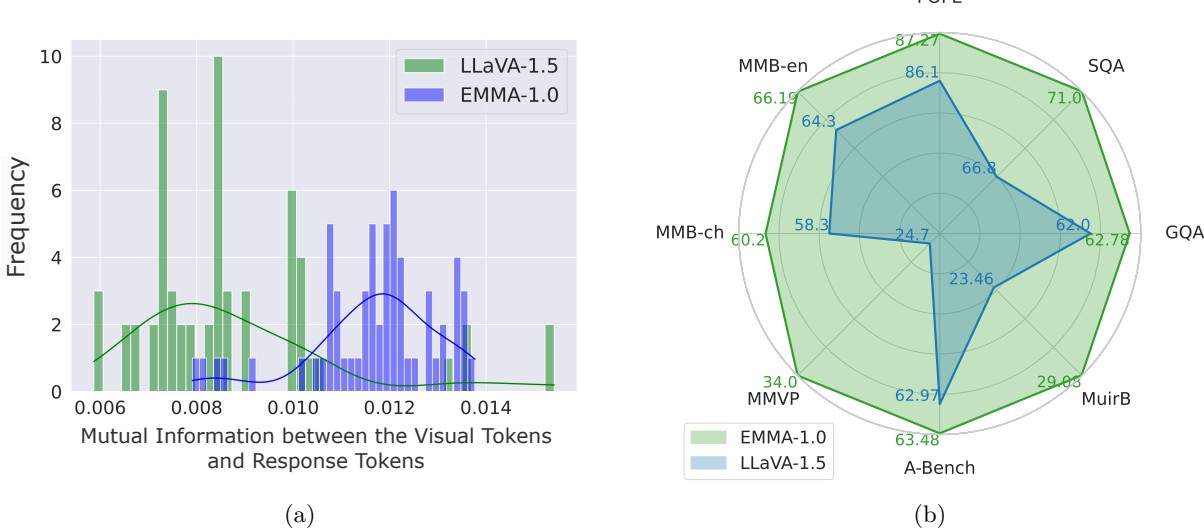

(a)                                                                                 (b)

Figure 3: **Left**: Figure 3a illustrates the mutual information between visual representations and response encodings. EMMA's visual representations exhibit $1.5\times$ higher mutual information with the response encodings compared to LLaVA's, highlighting its superior alignment with the instruction. **Right**: Figure 3b isolates the impact of EMMA's modality adaptation module by comparing it to the baseline model, LLaVA-1.5, while keeping all other parameters constant except for the architectural innovation.

visual tokens $\tilde{\mathbf{v}}$. The complete pipeline can be expressed as follows:

$$\tilde{\mathbf{v}} = f(v(\mathbf{x}),\ p(t(\mathbf{i}))) \tag{1}$$

Note that the dimensions of this alignment layer are designed to maintain the same number of visual tokens as the baseline model, ensuring consistency in the number of visual tokens passed to the language model. The Visual Alignment module plays a critical role in ensuring effective alignment between visual and instructions encodings, highlighting the most relevant tokens from the visual encodings in response to the instruction, and thereby delivering more precise visual information to the language model. Its lightweight design facilitates (1) easier interpretability and analysis, and more importantly, (2) its simplicity mitigates overfitting when training on small datasets and (3) reduces the training and inference time while outperforming the state-of-the-art models with $10\times$ fewer parameters. Moreover, by leveraging the inherent alignment of CLIP's vision and text encoders, both encoders remain frozen during training. This prevents overfitting to the training data and preserves the generalization capabilities of the original encoders, enabling EMMA to deliver robust performance across diverse tasks.

### 3.2 Analysis on Modality Adaptation by EMMA

In this section, we analyze the impact of the modality adaptation module introduced by EMMA.

**The Mechanics of the Visual Alignment Module.** The Visual Alignment module takes the concatenation of the visual and textual tokens ($n+m$ tokens) as input to generate the $n$ refined visual tokens. We begin our examination by scrutinizing the matrix $W \in \mathbb{R}^{(n+m)\times n}$ associated with it. By analyzing the norms of the weights corresponding to each token, we can identify which tokens, visual or textual, are most impactful. The histogram of $\ell_1$ norm of visual and textual tokens are demonstrated in Figure 2. As the weights of textual tokens are below 1 and the weights of visual tokens are above 1, the $\ell_1$ norm of the weights serve as a more indicative measure of token importance. As expected, the visual tokens exhibit higher weights within the Visual Alignment module, signifying their greater influence on the generated multi-modal representation. Another key observation is that certain textual tokens exert more influence than others. To highlight the relative importance of each textual token, Figure 2b presents a bar chart illustrating their respective

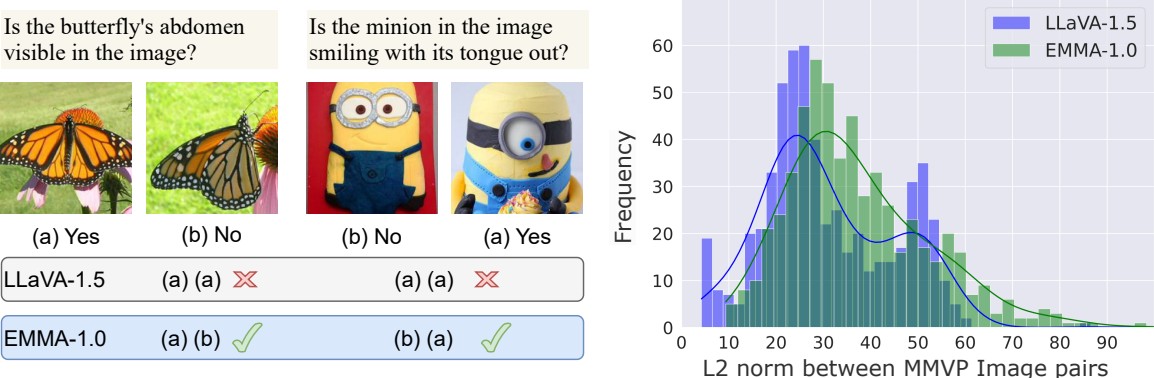

(a) Instances of MMVP Benchmark

(b) $\ell_2$ distance between the MMVP image pairs.

Figure 4: **Is EMMA capable of discerning images by focusing on the aspect specified in the instruction?** To address this, we utilize the MMVP benchmark, which contains closely resembling images, as shown in this figure. By leveraging EMMA's visual representations of the image pairs and calculating the $\ell_2$ norm, we observe an increase in the distance between them, as illustrated in Figure 4b, indicating EMMA's ability to differentiate between similar images.

weights. CLIP's text encoder generates 77 textual tokens, shown along the x-axis. As depicted in the figure, the earlier tokens tend to have a more significant impact. This finding indicates that the alignment module effectively identifies the most informative textual tokens, as instructions typically consist of brief prompts, with the crucial information concentrated in the early tokens and the remaining tokens often masked.

**Mutual Information between Aligned Visual Tokens and Response Tokens.** Another primary objective of the Modality Adaptation module is to adapt the visual representations to the language modality, ensuring they are well-aligned with the language model and encapsulate the necessary information to accurately respond to the given prompt. To evaluate the contribution of visual tokens to the language model's output, we use the Mutual Information, which quantifies the amount of information obtained about one random variable through the other. In this analysis, the LLaVA-In-Wild Liu et al. (2024b) benchmark is employed, which has a set of 24 images with 60 challenging questions in novel domains. For each of the 60 samples, visual representations are generated using the visual modules of both LLaVA and EMMA. It is important to note that EMMA's visual module processes the image and prompt to produce instruction-aware representations, whereas LLaVA generates instruction-agnostic encodings. Additionally, for each sample, the corresponding answer is encoded using CLIP's text encoder. The Mutual Information between the visual and response encodings is then calculated, with the results shown in Figure 3a. As illustrated, the mean mutual information for EMMA is 1.5 times higher than that of LLaVA, underscoring the effectiveness of EMMA's Visual Alignment in steering the model toward accurate language responses.

**Isolated Impact of the Modality Adaptation Module.** To evaluate the improvements introduced by EMMA's modality adaptation, we follow the same training scheme as the baseline model (Liu et al., 2024a). Therefore the improvements demonstrated in Figure 3b are achieved solely by our proposed method, EMMA. Across multiple benchmarks compared to the baseline model, LLaVA-1.5. EMMA consistently surpasses LLaVA-1.5 in various cross-modal tasks, demonstrating significant improvements in MMVP (Tong et al., 2024) (+9.3%), MuirBench (Wang et al., 2024a) (+5.62%), SQA (Lu et al., 2022b) (+4.2%), MM-Bench (Liu et al., 2023a) (+1.9%), and POPE (Li et al., 2023e) (+1.17%). These results underscore EMMA's effectiveness in enhancing cross-modal tasks that require both visual and textual comprehension.

**Aligned vs. Vanilla Visual Tokens** The primary objective of EMMA's modality adaptation is to align visual representations with the instructions, ensuring they emphasize the aspects of the image directed by the instructions. Our method achieves this by incorporating instruction encodings into the refinement process of the visual representations. In this section, we explore the alignment capabilities of EMMA by examining

| Method | LLM | VE | #Params | PT + IT | VQA$^{v2}$ | VisWiz | SQA$^{I}$ | GQA | OkVQA |
|--------|-----|-----|---------|---------|--------|--------|------|-----|-------|
| **BLIP-2** | Vicuna-13B | ViT-g/14 | 13B | 129M | 65.0 | 19.6 | 61 | 41 | 45.9 |
| **InstructBLIP** | Vicuna-7B | ViT-g/14 | 7.91B | 130.2M | - | 34.5 | 60.5 | 49.2 | - |
| **InstructBLIP** | Vicuna-13B | ViT-g/14 | 14.2B | 130.2M | - | 33.4 | 63.1 | 49.5 | - |
| **Shikra** | Vicuna-13B | ViT-L/14 | 13b | 6.1M | 77.4 | - | - | - | 47.2 |
| **IDEFICS** | LLaMA-7B | ViT-H/14 | 9B | 354M | 50.9 | 35.5 | - | 38.4 | - |
| **IDEFICS** | LLaMA-65B | ViT-H/14 | 80B | 354M | 60.0 | 36.0 | - | 45.2 | - |
| **Qwen-VL-Chat** | Qwen-7B | ViT-G/14 | 9.6B | 1.4B | 78.2 | 38.9 | 68.2 | 57 | 56.6 |
| **LLaVA-1.5** | Vicuna-7B | ViT-L/14 | 7B | 1.2M | 78.5 | 50.0 | 66.8 | **62.0** | - |
| **QA-ViT** | Vicuna-7B | ViT-L/14 | 7B | 1.2M | 80.5 | 36.5 | - | - | - |
| **mPLUG-Owl2** | LLaMA-7B | ViT-L/14 | 8.2B | 348M | 79.4 | 54.5 | 68.7 | 56.1 | 57.7 |
| **BRAVE** | FlanT5-XL | ♣ | 10B | 100M | 82.5 | 54.2 | - | 52.7 | 66.0 |
| **EMMA** | Vicuna-7B | ViT-L/14 | 7B | 1.8M | **89.42** | **56.03** | **73.14** | 56.01 | **68.57** |

Table 2: Comparison between EMMA and previous methods on academic task-oriented datasets. EMMA achieves state-of-the-art performance on 4/5 of the benchmarks while using the fewest number of parameters and training data compared to the previous approaches, utilizing modality adaptation(PT and IT stand for Pretraining and Instruction fine-tuning, respectively). The ♣ symbol denotes the collection of vision encoders utilized by BRAVE, including EVA-CLIP-g Fang et al. (2023), ViT-L/14 Radford et al. (2021), SILC-G/16 Naeem et al. (2025), ViT-e Chen et al. (2022), and DINOv2-L/14 Oquab et al. (2023).

the visual representations before and after alignment. To perform this analysis, we utilize the MMVP Tong et al. (2024) benchmark, which is designed to expose the visual shortcomings of MLLMs. We focus on images with visually similar encodings but subtle differences, as shown in Figure 4a. EMMA demonstrates a 9.3% improvement on this benchmark, underscoring its ability to generate more distinguishable visual representations for such images. To empirically validate this, we compute the $\ell_2$ norm of visual representations between randomly selected pairs of MMVP images, comparing the results for pre-aligned and post-aligned representations. The histogram of the norms, shown in Figure 4b, reveals a clear shift, indicating that the aligned representations are better at distinguishing between these images by focusing on instruction-relevant tokens.

## 4 Experimental Evaluation

In this section, we begin by comparing EMMA with state-of-the-art multi-modal LLMs, using the benchmarks introduced earlier. Following this, we conduct a robustness analysis with a focus on hallucination. We conduct an ablation study to identify the optimal layer output from the text encoder for use as textual representations.

**Implementation Details.** For training, we follow the same two-stage instruction fine-tuning process as LLaVA. In the pretraining stage, only the Visual Alignment and Projection modules are trained, while the language model remains frozen. During the fine-tuning stage, the LLM is unfrozen and fine-tuned along with the two aforementioned modules. We employ CLIP-ViT-L-14, trained on $336^2$ pixel images, as the base image encoder and text encoder. The Visual Alignment module is initialized with the identity matrix for the visual tokens and all zeros for the instruction tokens to transfer all the visual tokens at the beginning of training. Moreover, the Visual Alignment module is designed to maintain the same number of visual tokens as the baseline model. The latest Vicuna v1.5 (Zheng et al., 2023) is used as the base LLM. EMMA uses the same set of hyperparameters as the LLaVA-1.5. For all the analysis performed in the Section 3, we use the same dataset as the baseline model, which is 558K and 665K samples for the pretraining and fine-tuning stages, respectively. In the Evaluation setting, we have preserved the same pertaining data but scaled the fine-tuning data to 1.2M samples, including LVIS-Instruct4V(Wang et al., 2023a), CLEVR(Johnson et al., 2017), VizWiz(Gurari et al., 2018), and ScienceQA(Lu et al., 2022a) training data. EMMA's training data is the most efficient compared to all state-of-the-art methods (except for LLaVA), as shown in Table 3.

**Benchmarks.** We evaluate EMMA across 10 benchmarks that span a diverse set of tasks, including scientific question answering, visual question answering with poor image quality, integrated perception and reasoning tasks, visual dialogue, and general reasoning. The results of recent benchmarks designed specifically for

| Method | LLM | VE | #Params | PT + IT | MMB$^{EN}$ | MMB$^{CN}$ | MMMU | MathVista | Muirbench |
|---|---|---|---|---|---|---|---|---|---|
| **BLIP-2** | Vicuna-13B | ViT-g/14 | 14.2B | 129M | - | - | - | - | - |
| **InstructBLIP** | Vicuna-7B | ViT-g/14 | 7.91B | 130.2M | 36 | 23.7 | - | - | - |
| **InstructBLIP** | Vicuna-13B | ViT-g/14 | 14.2B | 130.2M | - | - | - | **25.3** | - |
| **Shikra** | Vicuna-13B | ViT-L/14 | 13B | 6.1M | 58.8 | - | - | - | - |
| **IDEFICS** | LLaMA-7B | ViT-H/14 | 9B | 354M | 48.2 | 25.2 | - | 19.8 | - |
| **IDEFICS** | LLaMA-65B | ViT-H/14 | 80B | 354M | 54.5 | 38.1 | - | - | - |
| **Qwen-VL-Chat** | Qwen-7B | ViT-G/14 | 9.6B | 1.4B | 60.6 | 56.7 | **35.9** | - | - |
| **LLaVA-1.5** | Vicuna-7B | ViT-L/14 | 7B | 1.2M | 64.3 | 58.3 | 35.11 | 21.1 | 23.46 |
| **mPLUG-Owl2** | LLaMA-7B | ViT-L/14 | 8.2B | 348M | 64.5 | - | 32.7 | 22.2 | - |
| **BRAVE** | FlanT5-XL | ♣ | 10B | 100M | - | - | - | - | |
| **EMMA** | Vicuna-7B | ViT-L/14 | 7B | 1.8M | **66.44** | **60.15** | 35.44 | 25.1 | **32.0** |

Table 3: This table compares EMMA with previous methods on specialized MLLM benchmarks. EMMA delivers the best performance on 3/5 benchmarks and secures second place on the remaining two with less than 0.5% difference. The ♣ symbol denotes the collection of vision encoders utilized by BRAVE, including EVA-CLIP-g Fang et al. (2023), ViT-L/14 Radford et al. (2021), SILC-G/16 Naeem et al. (2025), ViT-e Chen et al. (2022), and DINOv2-L/14 Oquab et al. (2023).

instruction-following large multi-modal models (LMMs) are presented in Table 3, while Table 2 outlines the performance on academic task-oriented benchmarks. EMMA emerges as the state-of-the-art model for 4 out of 5 academic task-oriented benchmarks, outperforming mPLUG-Owl2 which has 50× larger modality adaptation module and is trained on 300× more data, and BRAVE which has 24× larger vision encoder and is trained on 100× more data. On MLLM specialized benchmarks, EMMA delivers the best performance on 3 out of 5 benchmarks and is second best on the two others with less than 0.5% difference. In conclusion, EMMA achieves the best performance in 7 out of 10 benchmarks, despite utilizing the simplest architecture, outperforming other MLLMs that rely on complex modality adaptation modules.

**Robustness & Hallucination.** Robustness and the ability to avoid hallucination are essential for multi-modal large language models (MLLMs), which are increasingly applied in areas like medical diagnostics to interpret complex text and image data. Hallucination, a significant security threat to MLLMs, occurs when the model generates information that does not accurately represent the provided images or text. As a result, evaluating and mitigating hallucinations is a critical step in ensuring the reliability of MLLMs before real-world deployment, making it a key focus in model performance assessments. The hallucination evaluations in this section are performed utilizing two benchmarks, AMBER and FOIL, which do not rely on additional LLMs, thereby providing a direct and controlled means of assessing the model's ability to avoid hallucinations. These benchmarks (Wang et al., 2023b; Shekhar et al., 2017; Nie et al., 2024; Wu et al., 2024) focus on specific challenges in multi-modal reasoning, testing the model's accuracy in aligning textual and visual content without introducing erroneous information. AMBER consists of 7628, 4924, and 1663 samples for the Attribute, Relation, and Existence categories, respectively. FOIL contains a total of 99,480 test samples, of which 92,705 are straightforward, allowing both LLaVA and our method to successfully avoid hallucinations. However, 6,775 samples present more challenging cases. We compare LLaVA-1.5, the baseline for our approach, with EMMA. The results, shown in Figure 5a, indicate that EMMA outperforms the baseline, with significant performance gaps in two of the four benchmarks.

**Ablations on Text Encoder.** We ablate the text feature's abstraction level in this section. The textual features can be optionally derived from either the final layer or the penultimate layer of the CLIP Text Encoder. A comparison between the two methods of extracting textual features is presented in Figure 5b, demonstrating the clear superiority of features derived from the penultimate layer. We hypothesize that the final layer of CLIP captures more global and abstract semantics of the instruction, whereas the penultimate layer focuses on finer details. Additionally, since visual features are extracted from the corresponding layer in the visual encoder, using the same layer in the text encoder ensures both modalities operate at a similar level of abstraction, promoting better alignment between them.

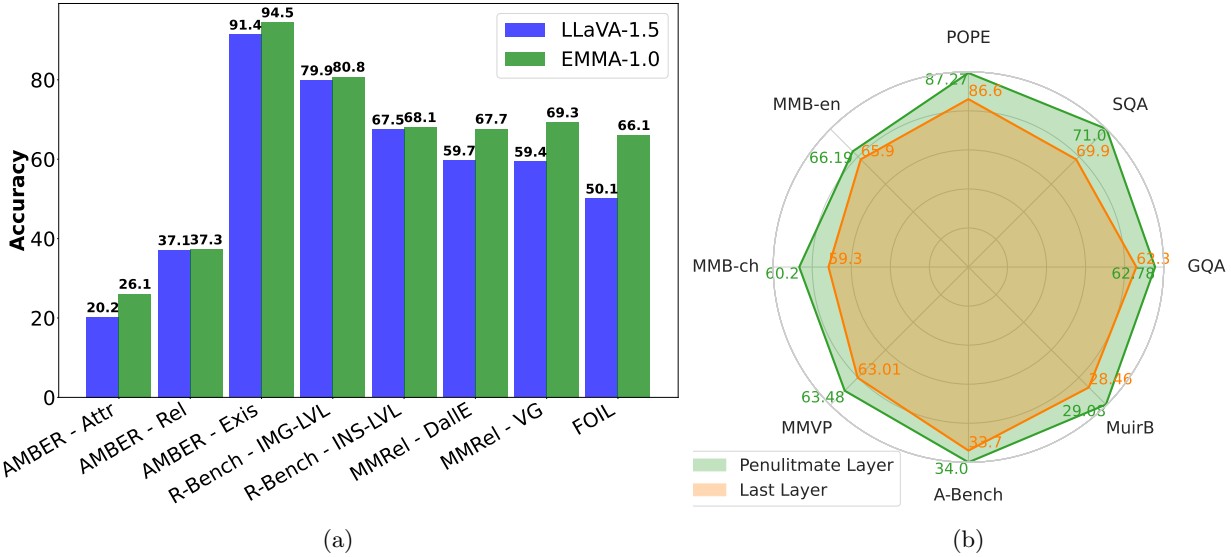

(a)  (b)

Figure 5: **Left**: This plot compares EMMA's robustness against hallucinations with LLaVA, demonstrating consistent improvements over the baseline in avoiding hallucinations. **Right**: This radar chart compares EMMA's performance when leveraging textual features generated by either the penultimate or last layer of the text encoder, highlighting the advantage of features associated with the penultimate layer.

**Ablation on the Visual Alignment Module.** To evaluate the effectiveness of our proposed visual alignment module, implemented as a linear layer, we conduct a comparative analysis by replacing the linear layer with a cross-attention mechanism between the two modalities. Specifically, we compare the performance of three MLLMs, all sharing the same architecture and training setup but differing in their modality adaptation modules:

- LLaVA – a baseline model without any modality adaptation,

- EMMA (Linear) – utilizing a linear layer for modality adaptation,

- EMMA (Cross-Attention) – using cross-attention for modality adaptation.

The results in Table 4 show that EMMA with the linear adaptation module consistently achieves the best performance across most benchmarks. Moreover, EMMA with cross-attention shows improvements over the baseline on SQA and MuirB, highlighting the potential of its adaptation mechanism. Nevertheless, its more complex architecture introduces risks of overfitting, as observed in benchmarks such as GCA, MMB, MMVP, and A-Bench. We posit that effective modality adaptation, i.e., refining visual representations through instruction-based guidance, requires a carefully balanced interplay between architectural complexity and training set size. This balance becomes particularly crucial when operating under constraints such as our baseline setup with only 1.2 million training samples.

| Method | Data | SQA | GQA | POPE | MMB-en | MMB-cn | MMVP | A-Bench | MuirB |
|---|---|---|---|---|---|---|---|---|---|
| LLaVA-1.5 | 1.2M | 66.80 | 62.00 | 86.10 | 64.30 | 58.30 | 24.70 | 62.97 | 23.46 |
| EMMA (Linear) | 1.2M | **71.00** | **62.78** | **87.27** | **66.19** | **60.20** | **34.00** | **63.48** | 29.08 |
| EMMA (Cross-Attention) | 1.2M | 67.08 | 49.33 | 79.73 | 60.84 | 51.42 | 8.67 | 56.01 | **29.50** |

Table 4: A comparative analysis of EMMA's Visual Alignment module reveals that the linear adaptation layer delivers the most consistent performance gains. In contrast, the cross-attention variant demonstrates selective improvements but is prone to potential overfitting in certain scenarios.

## 5 Conclusion

In this work, we addressed the inefficiencies in the modality adaptation modules employed by current multi-modal large language models. We hypothesized that the initial alignment between visual and textual encodings plays a critical role in determining the complexity level of the modality adaptation module and the necessary amount of training data. Our lightweight approach, EMMA (Enhanced Multi-Modal Adaptation), leverages CLIP's text encoder to generate instruction encodings and, by exploiting this initial alignment, demonstrates that the modality adaptation module can be simple while still enhancing the alignment between visual and textual modalities. Through extensive analysis, we demonstrated that EMMA effectively produces instruction-aware visual representations aligned with the language model. Our experiments, evaluated across multiple benchmarks, show that EMMA significantly outperforms state-of-the-art models that use modality adaptation modules 50× larger. Finally, our robustness analysis, particularly in hallucination avoidance, confirmed EMMA's superior ability to accurately process multi-modal data, even in challenging scenarios.

## Acknowledgments

This paper is supported in part by the Army Research Office under grant number W911NF-21-1- 0155 and by the New York University Abu Dhabi (NYUAD) Center for Artificial Intelligence and Robotics, funded by Tamkeen under the NYUAD Research Institute Award CG010. Additional support was provided by the NYU IT High Performance Computing resources, services, and staff expertise.

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
