# OpenReview forum: "EMMA: Efficient Visual Alignment in Multi-Modal LLMs"
_TMLR — Accepted by TMLR_

### Review · Reviewer_tZkv · 2025-03-31

**Summary Of Contributions:**

The paper introduces EMMA, a light module to enhance visual-textual alignment in multi-modal large language models. Current approaches prefer to use advanced modality adaptation modules, resulting in growing model size and large training data demands. EMMA addresses this by leveraging CLIP's pre-aligned vision and text encoders with a sparse linear layer for early fusion. This strategy maintains minimal parameter growth while boosting task performance. Comparison results show that EMMA outperforms state-of-the-art models on general and specialized tasks by up to 9.3%.

**Audience:**

Yes

**Claims And Evidence:**

Yes

**Requested Changes:**

Please check the weakness part.

**Strengths And Weaknesses:**

1.	The structure of EMMA is based on CLIP's joint training of text and vision encoders. The paper must address whether the approach generalizes across other vision-language architectures (e.g., DINOv2, PaLI) or performance is extremely CLIP-specific alignment-dependent.
2.	Experiments are done only with 7B-parameter LLMs. The paper does not state whether EMMA's efficiency holds for larger models (e.g., 70B+ LLMs) or whether performance gains degrade with size, limiting its generality.
3.	Though EMMA has less training data, comparison to models like mPLUG-Owl2 (trained from 300× more data) may not fully represent the impact of architectural efficiency. The paper should report whether baselines were re-trained under similar data constraints or whether findings are due to the intrinsic virtues of EMMA's architecture.
4.	Closer examination, e.g., attention pattern visualization or ablation on token importance, would lend stronger support to claims on alignment dynamics.

Remark

The VE and parameters column in Table 2 and Table 3 seems to be inaccurate for EMMA.

---

> ### Author Response · Authors · 2025-04-14
> **Response to Reviewer tZkv**
>
> Thank you for your thoughtful feedback. We’re glad you appreciated EMMA’s lightweight design and effectiveness. Below, we provide our answers to your questions and concerns.
>
>
> _**Question**: The structure of EMMA is based on CLIP's joint training of text and vision encoders. The paper must address whether the approach generalizes across other vision-language architectures (e.g., DINOv2, PaLI) or performance is extremely CLIP-specific alignment-dependent.
>
> **Answer**:The architecture LLaVA—where vision encodings are projected into the text space via a small number of linear layers and then concatenated with instruction tokens for input to the LLM—has become a standard paradigm for Multimodal Large Language Models (MLLMs), as seen in PaLM-E [3] and Shikra [4]. The recent InternVL2.5 [5] model also adopts this structure. This trend underscores the general applicability of our method, which can be seamlessly integrated into a wide range of MLLMs that follow this design.
>
> CLIP’s [1] visual encoder is a popular choice for encoding visual inputs, largely due to its pretraining on large-scale image-text pairs. The advantages of this alignment between visual and textual modalities have been recently explored [2]. EMMA builds upon this foundation by integrating CLIP’s text encoder with its visual encoder, enabling instruction-aware visual representation through an efficient modality adaptation module.
>
> _**Question**: Experiments are done only with 7B-parameter LLMs. The paper does not state whether EMMA's efficiency holds for larger models (e.g., 70B+ LLMs) or whether performance gains degrade with size, limiting its generality.
>
> **Answer**: EMMA is specifically proposed as an efficient adaptation method for resource-constrained settings, where computational and memory budgets are limited. As such, all of our experiments were conducted in this efficiency-first context using 7B-parameter models. While we have not performed experiments on 70B+ models due to resource limitations, the design of EMMA is in principle generalizable to larger models.
>
>
> _**Question**: Though EMMA has less training data, comparison to models like mPLUG-Owl2 (trained from 300× more data) may not fully represent the impact of architectural efficiency. The paper should report whether baselines were re-trained under similar data constraints or whether findings are due to the intrinsic virtues of EMMA's architecture.
>
> **Answer**: To ensure fair comparisons, we standardize the training setup—including datasets—across both LLaVA and EMMA. This controlled evaluation is detailed in Section 3.2 of the paper, specifically in the paragraph titled "Isolated Impact of the Modality Adaptation Module." The performance gains reported in Figure 3b are therefore based on an identical training configuration (LLaVA-1.5, Liu et al., 2024a), isolating the architectural contributions of EMMA, particularly the modality adaptation module, from any confounding factors such as data scale.
>
> As illustrated in Figure 3b, EMMA consistently outperforms LLaVA-1.5 across a variety of standard cross-modal benchmarks: MMVP (+9.3%, Tong et al., 2024), MuirBench (+5.62%, Wang et al., 2024a), SQA (+4.2%, Lu et al., 2022), MMBench (+1.9%, Liu et al., 2023a), and POPE (+1.17%, Li et al., 2023e). These results underscore the effectiveness of EMMA’s architecture in enhancing visual-textual reasoning, independent of training data size.
> Closer examination, e.g., attention pattern visualization or ablation on token importance, would lend stronger support to claims on alignment dynamics.
>
>
> _**Question**: The VE and parameters column in Table 2 and Table 3 seems to be inaccurate for EMMA.
>
> **Answer**:Thanks for pointing that out. Table 2 is correct; however, in Table 3, the 'VE' and '#Params' columns should be swapped for EMMA. We'll edit this in the revision.

---

### Review · Reviewer_fhZ6 · 2025-04-06

**Summary Of Contributions:**

EMMA proposes a lightweight visual-text alignment module that efficiently integrates CLIP’s pretrained vision and text encoders, which are inherently well aligned, significantly reducing computational overhead while maintaining strong cross-modal alignment. Through extensive interpretability analyses, EMMA demonstrates that textual embeddings effectively guide visual tokens, enhancing alignment quality.

**Audience:**

Yes

**Broader Impact Concerns:**

N/A — no additional concerns beyond what is discussed in the Broader Impact Statement.

**Claims And Evidence:**

Yes

**Requested Changes:**

1. **Clarify and Elaborate on the Visual Alignment Module**
* Please provide additional details on the Visual Alignment module, particularly how it handles variable-length visual and textual inputs. Including information on padding, masking, or aggregation strategies—as well as implementation-level details such as dimensions or initialization—would improve clarity and support reproducibility. A diagram could also be helpful for readers to better understand the architecture.
* Moreover, some technical details require clarification. For instance, in Section 3.1, the paper introduces an “Instruction Projection” module  p: \mathbb{R}^{d{\prime}} \rightarrow \mathbb{R}^{d}  to align the instruction embeddings with the visual token space. However, since both modalities are encoded using CLIP’s vision and text encoders—which are pretrained to produce outputs in the same dimensional space—this projection appears unnecessary. If  d = d{\prime}  by default in CLIP, the inclusion of this module is confusing and should be better justified or clarified.

2. **Provide More Information on Fine-Tuning Data Composition**
Since the paper reports strong performance gains, it would be helpful to include more information about the fine-tuning dataset, particularly how the 1.2M samples were constructed and what mixture of data sources were used. This would offer better context for understanding the results and allow others to reproduce the training setup more faithfully.

3. **Address Limitations for Multi-Turn Tasks**
EMMA appears to be optimized for single-turn tasks, and its applicability to multi-turn or conversational scenarios is unclear. Please discuss this limitation explicitly and consider outlining potential strategies or extensions that could support context-aware or multi-round reasoning.

**Strengths And Weaknesses:**

**strengths**

**1. Leveraging CLIP’s Pretrained Semantic Alignment.**
EMMA effectively leverages CLIP’s pretrained vision and text encoders, utilizing their inherent semantic alignment. The early fusion of visual and textual embeddings enhances semantic coherence efficiently, significantly reducing computational overhead.

**2. Comprehensive Interpretability Analysis.**
The paper provides insightful analyses (mutual information, weight norms), clearly showing distinct contributions from visual and textual embeddings. It demonstrates that textual instructions serve as subtle guides for visual tokens, supporting EMMA’s design rationale effectively.

**3. Lightweight and Efficient Alignment Module.**
EMMA introduces an exceptionally lightweight alignment module (~0.2% additional parameters), significantly lighter than methods like mPLUG-Owl2. Its simple linear aggregation notably reduces training complexity and inference costs, enhancing practical applicability.

**4. Strong Empirical Performance.**
EMMA achieves state-of-the-art performance across multiple benchmarks (e.g., VQA, ScienceQA, VizWiz), with improved robustness against hallucinations. However, I have concerns about its reported superiority, as critical experimental details (dataset composition, fine-tuning settings) lack sufficient clarity, raising fairness and reproducibility questions.

**weaknesses**

**1. Inherent Limitation to Single-Round Tasks**
EMMA’s lightweight alignment module is inherently limited to single-turn tasks, restricting its effectiveness in multi-turn conversational scenarios. This architectural limitation could inadvertently lead to shortcut learning or overfitting specifically to single-turn question-answer benchmarks. Since real-world multimodal applications frequently require multi-turn reasoning, context maintenance, and dialogue management, this significantly constrains EMMA’s broader applicability and impact in realistic, interactive settings.

**2. Unclear Explanation and Ambiguous Technical Details**
Technical details, particularly about the linear adaptation layer (Section 3.1), remain unclear. Crucial implementation specifics—such as handling variable-length tokens, padding, masking, or aggregation—are inadequately explained, complicating reproducibility and rigorous evaluation.

**3. Problematic Experimental Setting and Comparison Fairness**
EMMA nearly doubles its fine-tuning dataset (665K → 1.2M samples) compared to its baseline (LLaVA-1.5) but directly compares results without clearly highlighting this difference. Performance gains may thus reflect increased data rather than architectural advantages, and unclear dataset details further reduce empirical credibility and reproducibility.

**4. Limited Innovation Compared to Prior Work**
 Its core idea—integrating textual instructions into visual representations for alignment—bears strong resemblance to prior work such as _Question Aware Vision Transformer for Multimodal Reasoning_. Compared to that, EMMA offers relatively incremental innovation, primarily in terms of implementation simplification (e.g., replacing attention with a linear fusion layer) and utilizing the inherent alignment in CLIP model's visual and text modalities.

---

> ### Author Response · Authors · 2025-04-14
> **Response to Reviewer fhZ6**
>
> Thank you for the thoughtful and detailed review. We appreciate your recognition of EMMA’s efficient method and interpretability analysis. Below, we provide our answers to your questions and concerns.
>
> _**1.**:Inherent Limitation to Single-Round Tasks.
>
> **Answer**: Thank you for bringing up this question. We address it in two parts. First, the prompt is handled by the LLM independently of EMMA’s modality adaptation. This means that for multi-round tasks, the full prompt is passed to the LLM each time.Second, EMMA’s modality adaptation module is designed to handle multi-turn interactions by generating instruction-aware representations at each turn. More specifically, in a multi-round dialogue such as:
>
> Q1, A1, Q2, A2, …
>
> During each iteration i, the current query Qi is processed through the modality adaptation component to produce prompt-aware visual representations. This enables EMMA to dynamically adapt to the evolving context in a conversation, leveraging each prompt independently to generate appropriate multimodal responses. Therefore, EMMA is not strictly confined to single-turn tasks and can, in fact, contribute meaningfully to multi-turn reasoning scenarios.
>
> _**2.**: Unclear Explanation and Ambiguous Technical Details: Technical details, particularly about the linear adaptation layer (Section 3.1), remain unclear. Crucial implementation specifics—such as handling variable-length tokens, padding, masking, or aggregation—are inadequately explained, complicating reproducibility and rigorous evaluation.
>
> **Answer**: As described in the paper, we utilize the vision and text encoders from CLIP [1] to extract fixed-length token representations from images and text prompts, respectively. Since CLIP outputs a consistent number of tokens per modality, issues related to variable-length inputs, padding, and masking are not applicable at this stage.
> Following the extraction of visual and textual embeddings, we apply early fusion through a lightweight Visual Alignment Module. This module consists of a single linear projection layer that operates jointly on both sets of tokens. It aligns and integrates the visual and textual representations into a unified multi-modal embedding, which is then passed to the downstream components of the model.
>
> [1] Learning Transferable Visual Models From Natural Language Supervision
>
> _**3.**: Problematic Experimental Setting and Comparison Fairness EMMA nearly doubles its fine-tuning dataset (665K → 1.2M samples) compared to its baseline (LLaVA-1.5) but directly compares results without clearly highlighting this difference. Performance gains may thus reflect increased data rather than architectural advantages, and unclear dataset details further reduce empirical credibility and reproducibility.
>
>
> **Answer**: To ensure fair comparisons, we standardize the training setup—including datasets—across both LLaVA and EMMA. This controlled evaluation is detailed in Section 3.2 of the paper, specifically in the paragraph titled "Isolated Impact of the Modality Adaptation Module." The performance gains reported in Figure 3b are therefore based on an identical training configuration (LLaVA-1.5, Liu et al., 2024a), isolating the architectural contributions of EMMA, particularly the modality adaptation module, from any confounding factors such as data scale.
>
> As illustrated in Figure 3b, EMMA consistently outperforms LLaVA-1.5 across a variety of standard cross-modal benchmarks: MMVP (+9.3%, Tong et al., 2024), MuirBench (+5.62%, Wang et al., 2024a), SQA (+4.2%, Lu et al., 2022), MMBench (+1.9%, Liu et al., 2023a), and POPE (+1.17%, Li et al., 2023e). These results underscore the effectiveness of EMMA’s architecture in enhancing visual-textual reasoning, independent of training data size.

---

> ### Author Response · Authors · 2025-04-14
>
> _**4.**: Limited Innovation Compared to Prior Work  Its core idea—integrating textual instructions into visual representations for alignment—bears strong resemblance to prior work such as Question Aware Vision Transformer for Multimodal Reasoning. Compared to that, EMMA offers relatively incremental innovation, primarily in terms of implementation simplification (e.g., replacing attention with a linear fusion layer) and utilizing the inherent alignment in CLIP model's visual and text modalities.
>
> **Answer**: As you noted, EMMA introduces an efficient modality adaptation module that aligns visual and textual modalities, contributing meaningfully to the broader research direction of multimodal alignment. EMMA stands apart from prior work like the Question-Aware Vision Transformer (QA-ViT) in several key ways:
>
> - The QA-ViT paper [1] embeds question-awareness directly into the vision encoder, altering its internal architecture. In contrast, EMMA preserves the integrity of the pre-trained vision encoder and instead introduces a lightweight modality adaptation module that operates post-encoding. This design choice has a crucial benefit: by leaving the vision encoder untouched, EMMA retains the generalization capabilities of its visual representations. As a result, it leverages the strengths of pre-trained encoders while enabling robust performance across diverse tasks.
> - Additionally, EMMA achieves multimodal adaptation for visual representations through a simple linear fusion layer, avoiding the complexity and overhead of attention-heavy designs. In terms of model size, EMMA is highly efficient, adding only ~1M parameters, compared to ~236M in the smallest QA-ViT variant [1,2].
> - This disparity also affects data efficiency: QA-ViT requires approximately 8.5M pretraining examples, whereas EMMA achieves strong results with just 0.5M examples.
> Finally, on benchmark evaluations, EMMA outperforms QA-ViT across the benchmarks. Please refer to Table 2 in the paper for more details.
>
>
> [1] Question Aware Vision Transformer for Multimodal Reasoning
>
> [2] https://github.com/amazon-science/QA-ViT?tab=readme-ov-file
>
> _**Question**:Moreover, some technical details require clarification. For instance, in Section 3.1, the paper introduces an “Instruction Projection” module p: \mathbb{R}^{d{\prime}} \rightarrow \mathbb{R}^{d} to align the instruction embeddings with the visual token space. However, since both modalities are encoded using CLIP’s vision and text encoders—which are pretrained to produce outputs in the same dimensional space—this projection appears unnecessary. If d = d{\prime} by default in CLIP, the inclusion of this module is confusing and should be better justified or clarified.
>
> **Answer**:As mentioned in the CLIP [1] paper, they use a linear projection to map from each encoder’s representation to the multi-modal embedding space. We utilize the projection module for the same purpose and also to align the dimensions for visual and textual encodings that are then concatenated and passed through the visual alignment module.
>
>
> _**Question**:Provide More Information on Fine-Tuning Data Composition Since the paper reports strong performance gains, it would be helpful to include more information about the fine-tuning dataset, particularly how the 1.2M samples were constructed and what mixture of data sources were used. This would offer better context for understanding the results and allow others to reproduce the training setup more faithfully.
>
> **Answer**: Thank you for the helpful suggestion — we’ll incorporate it into the revision. For your reference, we use the same pre-training data as LLaVA. Our instruction fine-tuning data includes LLaVA's data plus the following datasets: LVIS-Instruct4V [1], CLEVR [2], VizWiz [3], and ScienceQA [4].
>
> [1] To See is to Believe: Prompting GPT-4V for Better Visual Instruction Tuning
>
> [2] CLEVR: A Diagnostic Dataset for Compositional Language and Elementary Visual Reasoning
>
> [3] VizWiz Grand Challenge: Answering Visual Questions from Blind People
>
> [4] Learn to Explain: Multimodal Reasoning via Thought Chains for Science Question Answering

---

### Review · Reviewer_5AWv · 2025-04-09

**Summary Of Contributions:**

This paper designs an **Efficient Multi-Modal Adaptation (EMMA)** architecture, which fuses the CLIP text encoder and visual encoder early on to achieve a cross-modal representation before injecting it into Llama.
We conduct extensive experiments to demonstrate the effectiveness of the EMMA structure, thoroughly analyze its various phenomena, and evaluate it on standard benchmarks, achieving significant performance improvements.

**Audience:**

Yes

**Broader Impact Concerns:**

I do not have any broader impact concerns regarding this paper.

**Claims And Evidence:**

Yes

**Requested Changes:**

I would have appreciated seeing more detailed ablation studies and further elaboration on implementation details.

**Strengths And Weaknesses:**

Advantages:

The Efficient Multi-Modal Adaptation (EMMA) architecture is highly intuitive. By leveraging CLIP’s text encoder to collaborate with its visual encoder, EMMA naturally generates text-guided visual features. Such features allow the visual modality to better highlight task-relevant regions when injected into the large language model (LLM).
This intuitive design leads to strong empirical performance, achieving significant improvements compared to standard baselines such as Llava 1.5.

Disadvantages:

However, the experimental evaluation presented in this paper is relatively simplistic. The chosen baseline, Llava 1.5, is somewhat outdated. Additionally, several critical design decisions of EMMA—such as the necessity of employing CLIP’s text encoder, the choice between cross-attention and self-attention mechanisms, or the potential benefits of additional optimization strategies and pretraining stages—lack comprehensive comparative experiments and explorations to clearly justify these design choices.

---

> ### Author Response · Authors · 2025-04-14
> **Response to Reviewer 5AWv**
>
> Thank you for your thoughtful feedback. We're glad you found EMMA's design intuitive and effective in leveraging CLIP’s encoders for text-guided visual features. Below, we provide our answers to your questions and concerns.
>
> _**Question**: The experimental evaluation presented in this paper is relatively simplistic. The chosen baseline, Llava 1.5, is somewhat outdated.
>
>
> **Answer**: As our baseline, we include 11 models, with QA-ViT [1] and BRAVE [2] being particularly relevant. Both were released shortly before EMMA and represent early efforts at early modality fusion.
> Moreover, the architecture LLaVA—where vision encodings are projected into the text space via a small number of linear layers and then concatenated with instruction tokens for input to the LLM—has become a standard paradigm for Multimodal Large Language Models (MLLMs), as seen in PaLM-E [3] and Shikra [4]. The recent InternVL2.5 [5] model also adopts this structure. This trend underscores the general applicability of our method, which can be seamlessly integrated into a wide range of MLLMs that follow this design.
>
> [1] Question Aware Vision Transformer for Multimodal Reasoning
>
> [2] BRAVE : Broadening the visual encoding of vision-language models
>
> [3] PaLM-E: An Embodied Multimodal Language Model
>
> [4] Shikra: Unleashing Multimodal LLM's Referential Dialogue Magic
>
> [5] Expanding Performance Boundaries of Open-Source Multimodal Models with Model, Data, and Test-Time Scaling
>
> _**Question**: several critical design decisions of EMMA—such as the necessity of employing CLIP’s text encoder, the choice between cross-attention and self-attention mechanisms or the potential benefits of additional optimization strategies and pretraining stages—lack comprehensive comparative experiments and explorations to clearly justify these design choices.
>
> **Answer**:  CLIP’s [1] visual encoder is a popular choice for encoding visual inputs, largely due to its pretraining on large-scale image-text pairs. The advantages of this alignment between visual and textual modalities have been more explored recently [2]. EMMA builds upon this foundation by integrating CLIP’s text encoder with its visual encoder, enabling instruction-aware visual representation through an efficient modality adaptation module.
>
> To evaluate the effectiveness of our proposed visual alignment module—implemented as a linear layer— with an attention mechanism, we conduct a comparative analysis by replacing the linear layer with a cross-attention mechanism between the two modalities. The results of this comparison are presented in the table below. Specifically, we evaluate three MLLMs, all sharing the same architecture and training setup but differing in their modality adaptation modules:
>
> - LLaVA – a baseline model without any modality adaptation,
> - EMMA (Linear) – using a linear layer for modality adaptation,
> - EMMA (Cross-Attention) – using cross-attention for modality adaptation.
>
> The results show that EMMA with the linear adaptation module consistently achieves the best performance across most benchmarks. However, EMMA with cross-attention outperforms on the MuirB benchmark and shows improvements over the baseline on SQA and MuirB, highlighting the potential of its adaptation mechanism. Nevertheless, its more complex architecture introduces risks of overfitting, as observed in benchmarks such as GCA, MMB, MMVP, and A-Bench.
> We hypothesize that effective modality adaptation—refining visual representations using instruction encodings—requires a carefully balanced design. This is especially critical when working with a limited training set of only 1.2 million samples, as in our baseline setup.
>
> | Method	| Data | SQA	| GQA | POPE | MMB-en |	MMB-cn | MMVP | A-Bench | MuirB	|
> | ----  | ---- | ---- | ---- | ---- | ---- | ---- | ---- | ---- | ---- |
> LLaVA-1.5 |1.2M  | 66.8| 62.0 | 86.1 | 64.3 | 58.3 | 24.7 | 62.97 | 23.46 |
> EMMA (Linear)| 1.2M | 71.00 | 62.78 | 87.27 | 66.19 | 60.2 | 34  | 63.48 | 29.08 |
> EMMA (Cross-Attention) | 1.2M | 67.08 |  49.33 |	79.73 | 60.84  | 51.42	| 8.67	| 56.01 |	 29.5 |
>
> [1] Learning Transferable Visual Models From Natural Language Supervision
>
> [2] EAGLE: EXPLORING THE DESIGN SPACE FOR MULTIMODAL LLMS WITH MIXTURE OF ENCODERS

---

### Review · Reviewer_P8Hw · 2025-04-13

**Summary Of Contributions:**

This paper introduces an approach to aligning visual and textual features. The authors suggest extracting instruction-aware visual features by integrating vision and text features before inputting the visual data into the LLM. Compared to previous methods, such as LLaVA, the proposed EMMA demonstrates superior vision-text alignment and achieves better results on benchmarks. Additionally, EMMA is more lightweight compared to other models like mPLUG-Owl2.

**Audience:**

Yes

**Claims And Evidence:**

Yes

**Requested Changes:**

See weakness

**Strengths And Weaknesses:**

## Strengths
1. The problem under study is intriguing and has consistently held significance in the field of vision-language models.
2. This paper presents a systematic approach to designing the alignment module, which is notably lightweight compared to other methods, such as mPLUG2-Owl.
3. In addition to conducting extensive experiments to demonstrate the effectiveness of the proposed module, the authors have also performed an in-depth analysis of the mechanisms driving EMMA's success.
4. The paper is well-written and easy to read.

## Weaknesses
1. In this paper, the authors consistently emphasize the importance of achieving better alignment between visual and textual features. However, from my perspective, the key to EMMA's success lies in its ability to extract instruction-aware visual features. This ensures that the features fed into the LLM are more closely related to the instructions, allowing the LLM to understand them more easily.
2. The implementation of EMMA is somewhat counterintuitive. Given that there is no direct interaction between the visual and textual features through a linear projection layer—since its role is merely to merge these features—it's puzzling why the authors did not opt for other approaches, such as cross-attention, to implement EMMA.
3. A question arises regarding alignment. In the literature, better alignment typically refers to the alignment between the space of the vision encoder and that of the LLM. However, EMMA only incorporates text features from CLIP's text encoder into the vision features, and the space of CLIP's text encoder is not aligned with that of the LLM. So how better alignment is achieved?
4. In Section 3.2, the authors assert that "instructions typically consist of brief prompts, with the crucial information concentrated in the early tokens." However, in reality, this is not always the case. Given that CLIP's text encoder can only process 77 tokens, how do the authors propose to address the issue of longer instructions, such as those that include few-shot examples or chain-of-thought (COT) reasoning?
5. I noticed that the authors only compare EMMA to methods that emerged about a year ago, and I encourage them to include comparisons with more recent works. Additionally, the training settings, such as the datasets used, are outdated. It is highly recommended that the authors update their training settings.

---

> ### Author Response · Authors · 2025-04-15
> **Response to Reviewer P8Hw**
>
> Thank you for your thoughtful feedback. We're glad you found the problem engaging and appreciated EMMA's lightweight modality adaptation module. Below, we provide our answers to your questions and concerns.
>
> **1.** In this paper, the authors consistently emphasize the importance of achieving better alignment between visual and textual features. However, from my perspective, the key to EMMA's success lies in its ability to extract instruction-aware visual features. This ensures that the features fed into the LLM are more closely related to the instructions, allowing the LLM to understand them more easily.
>
> **Answer**:  By visual alignment, we refer to the process of aligning visual representations with the given instruction, ensuring the visual representations are semantically closer to the elements the LLM needs to attend to. The result of this alignment is instruction-aware visual features—representations that are tailored to the context of the instruction.
>
> **2.** The implementation of EMMA is somewhat counterintuitive. Given that there is no direct interaction between the visual and textual features through a linear projection layer—since its role is merely to merge these features—it's puzzling why the authors did not opt for other approaches, such as cross-attention, to implement EMMA.
>
> **Answer**: To evaluate the effectiveness of our proposed visual alignment module—implemented as a linear layer with an attention mechanism—we conduct a comparative analysis by replacing the linear layer with a cross-attention mechanism between the two modalities. The results of this comparison are presented in the table below. Specifically, we evaluate three MLLMs, all sharing the same architecture and training setup but differing in their modality adaptation modules:
>
> - LLaVA – a baseline model without any modality adaptation,
> - EMMA (Linear) – using a linear layer for modality adaptation,
> - EMMA (Cross-Attention) – using cross-attention for modality adaptation.
>
> The results show that EMMA with the linear adaptation module consistently achieves the best performance across most benchmarks. However, EMMA with cross-attention outperforms on the MuirB benchmark and shows improvements over the baseline on SQA and MuirB, highlighting the potential of its adaptation mechanism. Nevertheless, its more complex architecture introduces risks of overfitting, as observed in benchmarks such as GCA, MMB, MMVP, and A-Bench.
> We hypothesize that effective modality adaptation—refining visual representations using instruction encodings—requires a carefully balanced design. This is especially critical when working with a limited training set of only 1.2 million samples, as in our baseline setup.
>
> | Method	| Data | SQA	| GQA | POPE | MMB-en |	MMB-cn | MMVP | A-Bench | MuirB	|
> | ----  | ---- | ---- | ---- | ---- | ---- | ---- | ---- | ---- | ---- |
> LLaVA-1.5 |1.2M  | 66.8| 62.0 | 86.1 | 64.3 | 58.3 | 24.7 | 62.97 | 23.46 |
> EMMA (Linear)| 1.2M | 71.00 | 62.78 | 87.27 | 66.19 | 60.2 | 34 | 63.48 | 29.08 |
> EMMA (Cross-Attention) | 1.2M | 67.08 |  49.33 |	79.73 | 60.84  | 51.42	| 8.67	| 56.01 |	 29.5 |
>
> **3.** A question arises regarding alignment. In the literature, better alignment typically refers to the alignment between the space of the vision encoder and that of the LLM. However, EMMA only incorporates text features from CLIP's text encoder into the vision features, and the space of CLIP's text encoder is not aligned with that of the LLM. So how better alignment is achieved?
>
> **Answer**: While CLIP’s encoder space isn’t explicitly aligned with language models, recent studies have shown that its encodings align more closely with LLMs than other vision encoders like DINOv2—largely due to its pretraining on large-scale image-text pairs [2]. Building on this foundation, EMMA integrates CLIP’s text and visual encoders, enabling instruction-aware visual representations through an efficient modality adaptation module.
>
> [1] Learning Transferable Visual Models From Natural Language Supervision
>
> [2] EAGLE: EXPLORING THE DESIGN SPACE FOR MULTIMODAL LLMS WITH MIXTURE OF ENCODERS

---

> > ### Author Response · Authors · 2025-04-15
> >
> > **4.** In Section 3.2, the authors assert that "instructions typically consist of brief prompts, with the crucial information concentrated in the early tokens." However, in reality, this is not always the case. Given that CLIP's text encoder can only process 77 tokens, how do the authors propose to address the issue of longer instructions, such as those that include few-shot examples or chain-of-thought (COT) reasoning?
> >
> > **Answer**: We address this concern by distinguishing between how prompts are handled by the LLM and how they are processed by EMMA’s modality adaptation module.
> > First, the full prompt—including long-form instructions, few-shot examples, or chain-of-thought reasoning—is handled entirely by the LLM. The complete context is passed to the LLM at each inference step, ensuring it has full access to all relevant information.
> > Second, EMMA’s modality adaptation module only requires the current query to generate instruction-aware visual representations. For example, in an in-context learning scenario with k example pairs and a final query:
> >
> > Q1, A1, Q2, A2, …, Qk, Ak, Qc
> >
> > Only the final query Qc is fed into the modality adaptation module to produce the visual features aligned with the current context. Similarly, in a multi-round dialogue setup:
> >
> > Q1, A1, Q2, A2, …, Qi
> >
> > At each round i, only the current question Qi is used by the adaptation module.
> >
> > During each iteration i, the current query Qi is processed through the modality adaptation component to produce prompt-aware visual representations. This enables EMMA to **dynamically adapt to the evolving context** in a conversation, leveraging each prompt independently to generate appropriate multimodal responses.
> >
> >
> > **5.** I noticed that the authors only compare EMMA to methods that emerged about a year ago, and I encourage them to include comparisons with more recent works. Additionally, the training settings, such as the datasets used, are outdated. It is highly recommended that the authors update their training settings.
> >
> >
> > **Answer**: In our current version, we include 11 baseline models, including QA-ViT [1] and BRAVE [2], which were released shortly before EMMA and are related works representing early efforts at early modality fusion.
> > Moreover, the architecture LLaVA—where vision encodings are projected into the text space via a small number of linear layers and then concatenated with instruction tokens for input to the LLM—has become a standard paradigm for Multimodal Large Language Models (MLLMs), as seen in PaLM-E [3] and Shikra [4]. The recent InternVL2.5 [5] model also adopts this structure. EMMA is well-aligned with this trend and can be readily integrated into such architectures, demonstrating its continued relevance and scalability.
> >
> >
> > [1] Question Aware Vision Transformer for Multimodal Reasoning
> >
> > [2] BRAVE : Broadening the visual encoding of vision-language models
> >
> > [3] PaLM-E: An Embodied Multimodal Language Model
> >
> > [4] Shikra: Unleashing Multimodal LLM's Referential Dialogue Magic
> >
> > [5] Expanding Performance Boundaries of Open-Source Multimodal Models with Model, Data, and Test-Time Scaling

---

### Author Response · Authors · 2025-04-15
**Revision of the paper**

Dear Reviewers,

Thank you for your constructive feedback. We have carefully revised the paper in response to your suggestions. The following changes have been made in the updated submission:

- We have included an ablation study in Section 4 that compares EMMA’s Linear Visual Alignment module with a cross-attention variant. The results of this analysis are summarized in Table 4.

- Details regarding the instruction fine-tuning data have been included in Section 4 of the paper.

- The VE column for EMMA in Table 3 has been corrected.

We appreciate your valuable input and hope the revisions address your concerns.

---

### Decision · Action_Editor_tL5W · 2025-05-26

**Recommendation:** Accept with minor revision

**Comment:**

This is a borderline submission with two weak accepts and two weak rejects. After carefully inspecting the paper, reviews and author feedback. AE recommends acceptance with minor revision because the benefits of the proposed approach outweigh the concerns pointed out by the reviewers in AE's view. In particular, the proposed approach is simple, intuitive, and demonstrates strong empirical performance on important and popular benchmarks. Regarding the minor revision, the AE recommends the authors to revisit the comments of reviewer P8Hw and incorporate parts of their responses into the method or even introduction section of the manuscript (e.g. aligning vision-text features vs making vision features instruction-aware, linear projection vs attention for combining CLIP vision/text features). Many of them are valid points that can benefit the readers. If possible (optional), adding some experiments (qualitative or quantitative) on how the proposed approach works with long-form instructions (e.g. few-shot, CoT) can enhance the value of this paper. Last but not least, given the strong performance of the proposed approach, it'd be very helpful for the community if authors can release the source code to aid other researchers reproduce the results reported in this paper.

**Audience:**

Yes, this paper should be of interest to TMLR audience. This is supported by all the reviewers.

**Claims And Evidence:**

Yes, the claims made in this submission were supported by concrete and clear evidence via comparison with other approaches and ablation studies. This is concurred by all the reviewers.